# Neural Variational Inference and Learning in Undirected Graphical Models

**Volodymyr Kuleshov**
Stanford University
Stanford, CA 94305
kuleshov@cs.stanford.edu

**Stefano Ermon**
Stanford University
Stanford, CA 94305
ermon@cs.stanford.edu

## Abstract

Many problems in machine learning are naturally expressed in the language of undirected graphical models. Here, we propose black-box learning and inference algorithms for undirected models that optimize a variational approximation to the log-likelihood of the model. Central to our approach is an upper bound on the log-partition function parametrized by a function $q$ that we express as a flexible neural network. Our bound makes it possible to track the partition function during learning, to speed-up sampling, and to train a broad class of hybrid directed/undirected models via a unified variational inference framework. We empirically demonstrate the effectiveness of our method on several popular generative modeling datasets.

## 1 Introduction

Many problems in machine learning are naturally expressed in the language of undirected graphical models. Undirected models are used in computer vision [1], speech recognition [2], social science [3], deep learning [4], and other fields. Many fundamental machine learning problems center on undirected models [5]; however, inference and learning in this class of distributions give rise to significant computational challenges.

Here, we attempt to tackle these challenges via new variational inference and learning techniques aimed at undirected probabilistic graphical models $p$. Central to our approach is an upper bound on the log-partition function of $p$ parametrized by a an approximating distribution $q$ that we express as a flexible neural network [6]. Our bound is tight when $q = p$ and is convex in the parameters of $q$ for interesting classes of $q$. Most interestingly, it leads to a lower bound on the log-likelihood function $\log p$, which enables us to fit undirected models in a variational framework similar to black-box variational inference [7].

Our approach offers a number of advantages over previous methods. First, it enables training undirected models in a black-box manner, i.e. we do not need to know the structure of the model to compute gradient estimators (e.g., as in Gibbs sampling); rather, our estimators only require evaluating a model's unnormalized probability. When optimized jointly over $q$ and $p$, our bound also offers a way to track the partition function during learning [8]. At inference-time, the learned approximating distribution $q$ may be used to speed-up sampling from the undirected model by initializing an MCMC chain (or it may itself provide samples). Furthermore, our approach naturally integrates with recent variational inference methods [6, 9] for directed graphical models. We anticipate that our approach will be most useful in automated probabilistic inference systems [10].

As a practical example for how our methods can be used, we study a broad class of hybrid directed/undirected models and show how they can be trained in a unified black-box neural variational inference framework. Hybrid models like the ones we consider have been popular in the early deep learning literature [4, 11] and take inspiration from the principles of neuroscience [12]. They also

possess a higher modeling capacity for the same number of variables; quite interestingly, we identify settings in which such models are also easier to train.

## 2 Background

**Undirected graphical models.** Undirected models form one of the two main classes of probabilistic graphical models [13]. Unlike directed Bayesian networks, they may express more compactly relationships between variables when the directionality of a relationship cannot be clearly defined (e.g., as in between neighboring image pixels).

In this paper, we mainly focus on Markov random fields (MRFs), a type of undirected model corresponding to a probability distribution of the form $p_\theta(x) = \tilde{p}_\theta(x)/Z(\theta)$, where $\tilde{p}_\theta(x) = \exp(\theta \cdot x)$ is an unnormalized probability (also known as energy function) with parameters $\theta$, and $Z(\theta) = \int \tilde{p}_\theta(x)dx$ is the partition function, which is essentially a normalizing constant. Our approach also admits natural extensions to conditional random field (CRF) undirected models.

**Importance sampling.** In general, the partition function of an MRF is often an intractable integral over $\tilde{p}(x)$. We may, however, rewrite it as

$$I := \int_x \tilde{p}_\theta(x)dx = \int_x \frac{\tilde{p}_\theta(x)}{q(x)}q(x)dx = \int_x w(x)q(x)dx, \tag{1}$$

where $q$ is a proposal distribution. Integral $I$ can in turn be approximated by a Monte-Carlo estimate $\hat{I} := \frac{1}{n}\sum_{i=1}^n w(x_i)$, where $x_i \sim q$. This approach, called *importance sampling* [14], may reduce the variance of an estimator and help compute intractable integrals. The variance of an importance sampling estimate $\hat{I}$ has a closed-form expression: $\frac{1}{n}\left(\mathbb{E}_{q(x)}[w(x)^2] - I^2\right)$. By Jensen's inequality, it equals 0 when $p = q$.

**Variational inference.** Inference in undirected models is often intractable. Variational approaches approximate this process by optimizing the evidence lower bound

$$\log Z(\theta) \geq \max_q \mathbb{E}_{q(x)}\left[\log \tilde{p}_\theta(x) - \log q(x)\right]$$

over a distribution $q(x)$; this amounts to finding a $q$ that approximates $p$ in terms of $KL(q||p)$. Ideal $q$'s should be expressive, easy to optimize over, and admit tractable inference procedures. Recent work has shown that neural network-based models possess many of these qualities [15, 16, 17].

**Auxiliary-variable deep generative models.** Several families of $q$ have been proposed to ensure that the approximating distribution is sufficiently flexible to fit $p$. This work makes use of a class of distributions $q(x, a) = q(x|a)q(a)$ that contain *auxiliary* variables $a$ [18, 19]; these are latent variables that make the marginal $q(x)$ multimodal, which in turn enables it to approximate more closely a multimodal target distribution $p(x)$.

## 3 Variational Bounds on the Partition Function

This section introduces a variational upper bound on the partition function of an undirected graphical model. We analyze its properties and discuss optimization strategies. In the next section, we use this bound as an objective for learning undirected models.

### 3.1 A Variational Upper Bound on $Z(\theta)$

We start with the simple observation that the variance of an importance sampling estimator (1) of the partition function naturally yields an upper bound on $Z(\theta)$:

$$\mathbb{E}_{q(x)}\left[\frac{\tilde{p}(x)^2}{q(x)^2}\right] \geq Z(\theta)^2. \tag{2}$$

As mentioned above, this bound is tight when $q = p$. Hence, it implies a natural algorithm for computing $Z(\theta)$: minimize (2) over $q$ in some family $\mathcal{Q}$.

We immediately want to emphasize that this algorithm will not be directly applicable to highly peaked and multimodal distributions $\tilde{p}$ (such as an Ising model near its critical point). If $q$ is initially very far from $\tilde{p}$, Monte Carlo estimates will tend to under-estimate the partition function.

However, in the context of learning $p$, we may expect a random initialization of $\tilde{p}$ to be approximately uniform; we may thus fit an initial $q$ to this well-behaved distribution, and as we gradually learn or anneal $p$, $q$ should be able to track $p$ and produce useful estimates of the gradients of $\tilde{p}$ and of $Z(\theta)$. Most importantly, these estimates are black-box and do not require knowing the structure of $\tilde{p}$ to compute. We will later confirm that our intuition is correct via experiments.

## 3.2 Properties of the Bound

**Convexity properties.** A notable feature of our objective is that if $q$ is an exponential family with parameters $\phi$, the bound is jointly log-convex in $\theta$ and $\phi$. This lends additional credibility to the bound as an optimization objective. If we choose to further parametrize $\phi$ by a neural net, the resulting non-convexity will originate solely from the network, and not from our choice of loss function.

To establish log-convexity, it suffices to look at $\tilde{p}_\theta(x)^2/q(x)$ for one $x$, since the sum of log-convex functions is log-convex. Note that $\log \frac{\tilde{p}_\theta(x)^2}{q(x)} = 2\theta^T x - \log q_\phi(x)$. One can easily check that a non-negative concave function is also log-concave; since $q$ is in the exponential family, the second term is convex, and our claim follows.

**Importance sampling.** Minimizing the bound on $Z(\theta)$ may be seen as a form of adaptive importance sampling, where the proposal distribution $q$ is gradually adjusted as more samples are taken [14, 20]. This provides another explanation for why we need $q \approx p$; note that when $q = p$, the variance is zero, and a single sample computes the partition function, demonstrating that the bound is indeed tight. This also suggests the possibility of taking $\frac{1}{n} \sum_{i=1}^{n} \frac{\tilde{p}(x_i)}{q(x_i)}$ as an estimate of the partition function, with the $x_i$ being all the samples that have been collected during the optimization of $q$.

$\chi^2$**-divergence minimization.** Observe that optimizing (2) is equivalent to minimizing $\mathbb{E}_q \frac{(\tilde{p}-q)^2}{q^2}$, which is the $\chi^2$-divergence, a type of $\alpha$-divergence with $\alpha = 2$ [21, 22]. This connections highlights the variational nature of our approach and potentially suggests generalizations to other divergences. Moreover, many interesting properties of the bound can be easily established from this interpretation, such as convexity in terms of $q, \tilde{p}$ (in functional space).

## 3.3 Auxiliary-Variable Approximating Distributions

A key part of our approach is the choice of approximating family $\mathcal{Q}$: it needs to be expressive, easy to optimize over, and admit tractable inference procedures. In particular, since $\tilde{p}(x)$ may be highly multi-modal and peaked, $q(x)$ should ideally be equally complex. Note that unlike earlier methods that parametrized conditional distributions $q(z|x)$ over hidden variables $z$ (e.g. variational autoencoders [15]), our setting does not admit a natural conditioning variable, making the task considerably more challenging.

Here, we propose to address these challenges via an approach based on *auxiliary-variable* approximations [18]: we introduce a set of latent variables $a$ into $q(x, a) = q(x|a)q(a)$ making the marginal $q(x)$ multi-modal. Computing the marginal $q(x)$ may no longer be tractable; we therefore apply the variational principle one more time and introduce an additional relaxation of the form

$$\mathbb{E}_{q(a,x)} \left[ \frac{p(a|x)^2 \tilde{p}(x)^2}{q(x|a)^2 q(a)^2} \right] \geq \mathbb{E}_{q(x)} \left[ \frac{\tilde{p}(x)^2}{q(x)^2} \right] \geq Z(\theta)^2, \quad (3)$$

where, $p(a|x)$ is a probability distribution over $a$ that lifts $\tilde{p}$ to the joint space of $(x, a)$. To establish the first inequality, observe that

$$\mathbb{E}_{q(a,x)} \left[ \frac{p(a|x)^2 \tilde{p}(x)^2}{q(x|a)^2 q(a)^2} \right] = \mathbb{E}_{q(x)q(a|x)} \left[ \frac{p(a|x)^2 \tilde{p}(x)^2}{q(a|x)^2 q(x)^2} \right] = \mathbb{E}_{q(x)} \left[ \frac{\tilde{p}(x)^2}{q(x)^2} \cdot \mathbb{E}_{q(a|x)} \left( \frac{p(a|x)^2}{q(a|x)^2} \right) \right].$$

The factor $\mathbb{E}_{q(a|x)} \left( \frac{p(a|x)^2}{q(a|x)^2} \right)$ is an instantiation of bound (2) for the distribution $p(a|x)$, and is therefore lower-bounded by 1.

This derivation also sheds light on the role of $p(a|x)$: it is an approximating distribution for the intractable posterior $q(a|x)$. When $p(a|x) = q(a|x)$, the first inequality in (3) is tight, and we are optimizing our initial bound.

### 3.3.1 Instantiations of the Auxiliary-Variable Framework

The above formulation is sufficiently general to encompass several different variational inference approaches. Either could be used to optimize our objective, although we focus on the latter, as it admits the most flexible approximators for $q(x)$.

**Non-parametric variational inference.** First, as suggested by Gershman et al. [23], we may take $q$ to be a uniform mixture of $K$ exponential families: $q(x) = \sum_{k=1}^{K} \frac{1}{K} q_k(x; \phi_k)$.

This is equivalent to letting $a$ be a categorical random variable with a fixed, uniform prior. The $q_k$ may be either Gaussians or Bernoulli, depending on whether $x$ is discrete or continuous. This choice of $q$ lets us potentially model arbitrarily complex $p$ given enough components. Note that for distributions of this form it is easy to compute the marginal $q(x)$ (for small $K$), and the bound in (3) may not be needed.

**MCMC-based variational inference.** Alternatively, we may set $q(a|x)$ to be an MCMC transition operator $T(x'|x)$ (or a sequence of operators) as in Salimans et al. [24]. The prior $q(a)$ may be set to a flexible distribution, such as normalizing flows [25] or another mixture distribution. This gives a distribution of the form

$$q(x, a) = T(x|a)q(a). \tag{4}$$

For example, if $T(x|a)$ is a Restricted Boltzmann Machine (RBM; Smolensky [26]), the Gibbs sampling operator $T(x'|x)$ has a closed form that can be used to compute importance samples. This is in contrast to vanilla Gibbs sampling, where there is no closed form density for weighting samples.

The above approach also has similarities to persistent contrastive divergence (PCD; Tieleman and Hinton [27]), a popular approach for training RBM models, in which samples are taken from a Gibbs chain that is not reset during learning. The distribution $q(a)$ may be thought of as a parametric way of representing a persistent distribution from which samples are taken throughout learning; like the PCD Gibbs chain, it too tracks the target probability $p$ during learning.

**Auxiliary-variable neural networks.** Lastly, we may also parametrize $q(a|x)$ by an flexible function approximator such as a neural network [18]. More concretely, we set $q(a)$ to a simple continuous prior (e.g. normal or uniform) and set $q_\phi(x|a)$ to an exponential family distribution whose natural parameters are parametrized by a neural net. For example, if $x$ is continuous, we may set $q(x|a) = N(\mu(a), \sigma(a)I)$, as in a variational auto-encoder. Since the marginal $q(x)$ is intractable, we use the variational bound (3) and parametrize the approximate posterior $p(a|x)$ with a neural network. For example, if $a \sim \mathcal{N}(0, 1)$, we may again set $p(a|x) = N(\mu(x), \sigma(x)I)$.

## 3.4 Optimization

In the rest of the paper, we focus on the auxiliary-variable neural network approach for optimizing bound (3). This approach affords us the greatest modeling flexibility and allows us to build on previous neural variational inference approaches.

The key challenge with this choice of representation is optimizing (3) with respect to the parameters $\phi, \phi$ of $p, q$. Here, we follow previous work on black-box variational inference [6, 7] and compute Monte Carlo estimates of the gradient of our neural network architecture.

The gradient with respect to $p$ has the form $2\mathbb{E}_q \frac{\tilde{p}(x,a)}{q(x,a)^2} \nabla_\phi \tilde{p}(x, a)$ and can be estimated directly via Monte Carlo. We use the score function estimator to compute the gradient of $q$, which can be written as $-\mathbb{E}_{q(x,a)} \frac{\tilde{p}(x,a)^2}{q(x,a)^2} \nabla_\phi \log q(x, a)$ and estimated again using Monte Carlo samples. In the case of a non-parametric variational approximation $\sum_{k=1}^{K} \frac{1}{K} q_k(x; \phi_k)$, the gradient has a simple expression $\nabla_{\phi_k} \mathbb{E}_q \frac{\tilde{p}(x)^2}{q(x)^2} = -\mathbb{E}_{q_k} \left[ \frac{\tilde{p}(x)^2}{q(x)^2} d_k(x) \right]$, where $d_k(x)$ is the difference of $x$ and its expectation under $q_k$.

Note also that if our goal is to compute the partition function, we may collect all intermediary samples for computing the gradient and use them as regular importance samples. This may be interpreted as a form of adaptive sampling [20].

**Variance reduction.** A well-known shortcoming of the score function gradient estimator is its high variance, which significantly slows down optimization. We follow previous work [6] and introduce two variance reduction techniques to mitigate this problem.

We first use a moving average $\bar{b}$ of $\tilde{p}(x)^2/q(x)^2$ to center the learning signal. This leads to a gradient estimate of the form $\mathbb{E}_{q(x)}(\frac{\tilde{p}(x)^2}{q(x)^2} - \bar{b})\nabla_\phi \log q(x)$; this yields the correct gradient by well known properties of the score function [7]. Furthermore, we use variance normalization, a form of adaptive step size. More specifically, we keep a running average $\bar{\sigma}^2$ of the variance of the $\tilde{p}(x)^2/q(x)^2$ and use a normalized form $g' = g/\max(1, \bar{\sigma}^2)$ of the original gradient $g$.

Note that unlike the standard evidence lower bound, we cannot define a sample-dependent baseline, as we are not conditioning on any sample. Likewise, many advanced gradient estimators [9] do not apply in our setting. Developing better variance reduction techniques for this setting is likely to help scale the method to larger datasets.

# 4 Neural Variational Learning of Undirected Models

Next, we turn our attention to the problem of learning the parameters of an MRF. Given data $\mathcal{D} = \{x^{(i)}\}_{i=1}^n$, our training objective is the log-likelihood

$$\log p(\mathcal{D}|\theta) := \frac{1}{n}\sum_{i=1}^n \log p_\theta(x^{(i)}) = \frac{1}{n}\sum_{i=1}^n \theta^T x^{(i)} - \log Z(\theta). \tag{5}$$

We can use our earlier bound to upper bound the log-partition function by $\log\left(\mathbb{E}_{x\sim q}\frac{\tilde{p}_\theta(x)^2}{q(x)^2}\right)$. By our previous discussion, this expression is convex in $\theta, \phi$ if $q$ is an exponential family distribution. The resulting lower bound on the log-likelihood may be optimized jointly over $\theta, \phi$; as discussed earlier, by training $p$ and $q$ jointly, the two distributions may help each other. In particular, we may start learning at an easy $\theta$ (where $p$ is not too peaked) and use $q$ to slowly track $p$, thus controlling the variance in the gradient.

**Linearizing the logarithm.** Since the log-likelihood contains the logarithm of the bound (2), our Monte Carlo samples will produce biased estimates of the gradient. We did not find this to pose problems in practice; however, to ensure unbiased gradients one may further linearize the log using the identity $\log(x) \leq ax - \log(a) - 1$, which is tight for $a = 1/x$. Together with our bound on the log-partition function, this yields

$$\log p(\mathcal{D}|\theta) \geq \max_{\theta, q} \frac{1}{n}\sum_{i=1}^n \theta^T x^{(i)} - \frac{1}{2}\left(a\mathbb{E}_{x\sim q}\frac{\tilde{p}_\theta(x)^2}{q_\psi(x)^2} - \log(a) - 1\right). \tag{6}$$

This expression is convex in each of $(\theta, \phi)$ and $a$, but is not jointly convex. However, it is straightforward to show that equation (6) and its unlinearized version have a unique point satisfying first-order stationarity conditions. This may be done by writing out the KKT conditions of both problems and using the fact that $a^* = (\mathbb{E}_{x\sim q}\frac{\tilde{p}_\theta(x)^2}{q(x)^2})^{-1}$ at the optimum. See Gopal and Yang [28] for more details.

## 4.1 Variational Inference and Learning in Hybrid Directed/Undirected Models

We apply our framework to a broad class of hybrid directed/undirected models and show how they can be trained in a unified variational inference framework.

The models we consider are best described as variational autoencoders with a Restricted Boltzmann Machine (RBM; Smolensky [26]) prior. More formally, they are latent-variable distributions of the form $p(x, z) = p(x|z)p(z)$, where $p(x|z)$ is an exponential family whose natural parameters are parametrized by a neural network as a function of $z$, and $p(z)$ is an RBM. The latter is an undirected latent variable model with hidden variables $h$ and unnormalized log-probability $\log \tilde{p}(z, h) = z^T W h + b^T z + c^T h$, where $W, b, c$ are parameters.

We train the model using two applications of the variational principle: first, we apply the standard evidence lower bound with an approximate posterior $r(z|x)$; then, we apply our lower bound on the RBM log-likelihood $\log p(z)$, which yields the objective

$$\log p(x) \geq \mathbb{E}_{r(z|x)} \left[ \log p(x|z) + \log \tilde{p}(z) + \log \mathcal{B}(\tilde{p}, q) - \log r(z|x) \right]. \tag{7}$$

Here, $\mathcal{B}$ denotes our bound (3) on the partition function of $p(z)$ parametrized with $q$. Equation (7) may be optimized using standard variational inference techniques; the terms $r(z|x)$ and $p(x|z)$ do not appear in $\mathcal{B}$ and their gradients may be estimated using REINFORCE and standard Monte Carlo, respectively. The gradients of $\tilde{p}(z)$ and $q(z)$ are obtained using methods described above. Note also that our approach naturally extends to models with multiple layers of latent directed variables.

Such hybrid models are similar in spirit to deep belief networks [11]. From a statistical point of view, a latent variable prior makes the model more flexible and allows it to better fit the data distribution. Such models may also learn structured feature representations: previous work has shown that undirected modules may learn classes of digits, while lower, directed layers may learn to represent finer variation [29]. Finally, undirected models like the ones we study are loosely inspired by the brain and have been studied from that perspective [12]. In particular, the undirected prior has been previously interpreted as an associative memory module [11].

# 5 Experiments

## 5.1 Tracking the Partition Function

We start with an experiment aimed at visualizing the importance of tracking the target distribution $p$ using $q$ during learning.

We use Equation 6 to optimize the likelihood of a $5 \times 5$ Ising MRF with coupling factor $J$ and unaries chosen randomly in $\{10^{-2}, -10^{-2}\}$. We set $J = -0.6$, sampled 1000 examples from the model, and fit another Ising model to this data. We followed a non-parametric inference approach with a mixture of $K = 8$

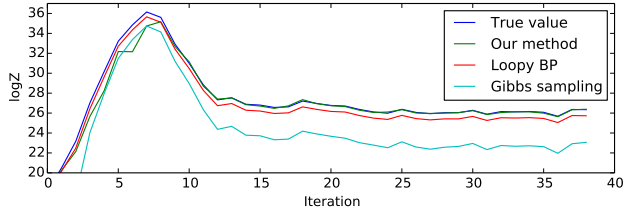

Bernoullis. We optimized (6) using SGD and alternated between ten steps over the $\phi_k$ and one step over $\theta, a$. We drew 100 Monte Carlo samples per $q_k$. Our method converged in about 25 steps over $\theta$. At each iteration we computed $\log Z$ via importance sampling.

The adjacent figure shows the evolution of $\log Z$ during learning. It also plots $\log Z$ computed by exact inference, loopy BP, and Gibbs sampling (using the same number of samples). Our method accurately tracks the partition function after about 10 iterations. In particular, our method fares better than the others when $J \approx -0.6$, which is when the Ising model is entering its phase transition.

## 5.2 Learning Restricted Boltzmann Machines

Next, we use our method to train Restricted Boltzmann Machines (RBMs) on the UCI digits dataset [30], which contains 10,992 $8 \times 8$ images of handwritten digits; we augment this data by moving each image 1px to the left, right, up, and down. We train an RBM with 100 hidden units using ADAM [31] with batch size 100, a learning rate of $3 \cdot 10^{-4}$, $\beta_1 = 0.9$, and $\beta_2 = 0.999$; we choose $q$ to be a uniform mixture of $K = 10$ Bernoulli distributions. We alternate between training $p$ and $q$, performing either 2 or 10 gradient steps on $q$ for each step on $p$ and taking 30 samples from $q$ per step; the gradients of $p$ are estimated via adaptive importance sampling.

We compare our method against persistent contrastive divergence (PCD; Tieleman and Hinton [27]), a standard method for training RBMs. The same ADAM settings were used to optimize the model with the PCD gradient. We used $k = 3$ Gibbs steps and 100 persistent chains. Both PCD and our method were implemented in Theano [32].

In Figure 1, we plot the true log-likelihood of the model (computed with annealed importance sampling with step size $10^{-3}$) as a function of the epoch; we use 10 gradient steps on $q$ for each step on $p$. Both PCD and our method achieve comparable performance. Interestingly, we may use our

Figure 1: Learning curves for an RBM trained with PCD-3 and with neural variational inference on the UCI digits dataset. Log-likelihood was computed using annealed importance sampling.

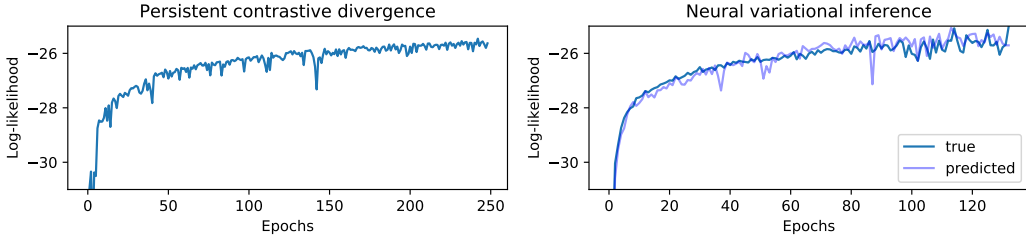

Table 1: Test set negative log likelihood on binarized MNIST and Omniglot for VAE and ADGM models with Bernoulli (200 vars) and RBM priors with 64 visible and either 8 or 64 hidden variables.

| Model | Binarized MNIST | | | Omniglot | | |
|---|---|---|---|---|---|---|
| | Ber(200) | RBM(64,8) | RBM(64,64) | Ber(200) | RBM(64,8) | RBM(64,64) |
| VAE | 111.9 | 105.4 | 102.3 | 135.1 | 130.2 | 128.5 |
| ADGM | 107.9 | 104.3 | 100.7 | 136.8 | 134.4 | 131.1 |

approximating distribution $q$ to estimate the log-likelihood via importance sampling. Figure 1 (right) shows that this estimate closely tracks the true log-likelihood; thus, users may periodically query the model for reasonably accurate estimates of the log-likelihood. In our implementation, neural variational inference was approximately eight times slower than PCD; when performing two gradient steps on $q$, our method was only 50% slower with similar samples and pseudo-likelihood; however log-likelihood estimates were noisier. Annealed importance sampling was always more than order of magnitude slower than neural variational inference.

**Visualizing the approximating distribution.** Next, we trained another RBM model performing two gradient steps for $q$ for each step of $p$. The adjacent figure shows the mean distribution of each component of the mixture of Bernoullis $q$; one may distinguish in them the shapes of various digits. This confirms that $q$ indeed approximates $p$.

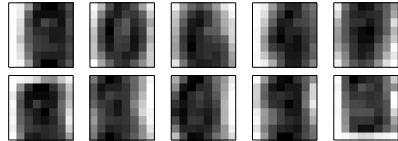

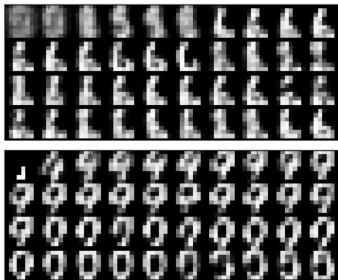

**Speeding up sampling from undirected models.** After the model has finished training, we can use the approximating $q$ to initialize an MCMC sampling chain. Since $q$ is a rough approximation of $p$, the resulting chain should mix faster. To confirm this intuition, we plot in the adjacent figure samples from a Gibbs sampling chain that has been initialized randomly (top), as well as from a chain that was initialized with a sample from $q$ (bottom). The latter method reaches a plausible-looking digit in a few steps, while the former produces blurry samples.

## 5.3 Learning Hybrid Directed/Undirected Models

Next, we use the variational objective (7) to learn two types of hybrid directed/undirected models: a variational autoencoder (VAE) and an auxiliary variable deep generative model (ADGM) [18]. We consider three types of priors: a standard set of 200 uniform Bernoulli variables, an RBM with 64 visible and 8 hidden units, and an RBM with 64 visible and 64 hidden units. In the ADGM, the approximate posterior $r(z, u|x) = r(z|u, x)r(u|x)$ includes auxiliary variables $u \in \mathbb{R}^{10}$. All the conditional probabilities $r(z|u, x), r(u|x), r(z|x), p(x|z)$ are parametrized with dense neural networks with one hidden layer of size 500.

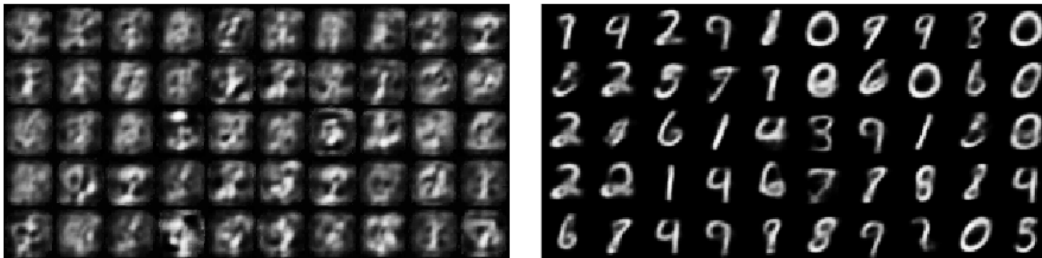

Figure 2: Samples from a deep generative model using different priors over the discrete latent variables $z$. On the left, the prior $p(z)$ is a Bernoulli distribution (200 vars); on the right, $p(z)$ is an RBM (64 visible and 8 hidden vars). All other parts of the model are held fixed.

We train all neural networks for 200 epochs with ADAM (same parameters as above) and neural variational inference (NVIL) with control variates as described in Mnih and Rezende [9]. We parametrize $q$ with a neural network mapping 10-dimensional auxiliary variables $a \in \mathcal{N}(0, I)$ to $x$ via one hidden layer of size 32. We show in Table 1 the test set negative log-likelihoods on the binarized MNIST [33] and $28 \times 28$ Omniglot [17] datasets; we compute these using $10^3$ Monte Carlo samples and using annealed importance sampling for the $64 \times 64$ RBM.

Overall, adding an RBM prior with as little as 8 latent variables results in significant log-likelihood improvements. Most interestingly, this prior greatly improves sample quality over the discrete latent variable VAE (Figure 2). Whereas the VAE failed to generate correct digits, replacing the prior with a small RBM resulted in smooth MNIST images. We note that both methods were trained with exactly the same gradient estimator (NVIL). We observed similar behavior for the ADGM model. This suggests that introducing the undirected component made the models more expressive and easier to train.

## 6   Related Work and Discussion

Our work is inspired by black-box variational inference [7] for variational autoencoders and related models [15], which involve fitting approximate posteriors parametrized by neural networks. Our work presents analogous methods for undirected models. Popular classes of undirected models include Restricted and Deep Boltzmann Machines [4, 26] as well as Deep Belief Networks [11]. Closest to our work is the discrete VAE model; however, Rolfe [29] seeks to efficiently optimize $p(x|z)$, while the RBM prior $p(z)$ is optimized using PCD; our work optimizes $p(x|z)$ using standard techniques and focuses on $p(z)$. Our bound has also been independently studied in directed models [22].

More generally, our work proposes an alternative to sampling-based learning methods; most variational methods for undirected models center on inference. Our approach scales to small and medium-sized datasets, and is most useful within hybrid directed-undirected generative models. It approaches the speed of the PCD method and offers additional benefits, such as partition function tracking and accelerated sampling. Most importantly, our algorithms are black-box, and do not require knowing the structure of the model to derive gradient or partition function estimators. We anticipate that our methods will be most useful in automated inference systems such as Edward [10].

The scalability of our approach is primarily limited by the high variance of the Monte Carlo estimates of the gradients and the partition function when $q$ does not fit $p$ sufficiently well. In practice, we found that simple metrics such as pseudo-likelihood were effective at diagnosing this problem. When training deep generative models with RBM priors, we noticed that weak $q$'s introduced mode collapse (but training would still converge). Increasing the complexity of $q$ and using more samples resolved these problems. Finally, we also found that the score function estimator of the gradient of $q$ does not scale well to higher dimensions. Better gradient estimators are likely to further improve our method.

## 7   Conclusion

In summary, we have proposed new variational learning and inference algorithms for undirected models that optimize an upper-bound on the partition function derived from the perspective of

importance sampling and $\chi_2$ divergence minimization. Our methods allow training undirected models in a black-box manner and will be useful in automated inference systems [10].

Our framework is competitive with sampling methods in terms of speed and offers additional benefits such as partition function tracking and accelerated sampling. Our approach can also be used to train hybrid directed/undirected models using a unified variational framework. Most interestingly, it makes generative models with discrete latent variables both more expressive and easier to train.

**Acknowledgements.** This work is supported by the Intel Corporation, Toyota, NSF (grants 1651565, 1649208, 1522054) and by the Future of Life Institute (grant 2016-158687).

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
