[Reviews · NeurIPS 2017]

Reviewer 1



In this paper the authors essentially propose to train a MLP to generate proposal samples which are used to estimate the partition function Z of an undirected model. Instead of using straight importance sampling to estimate Z (which would be an unbiased estimator for Z), they propose a bound that overestimates Z^2 *in expectation*. While the authors highlight around line 70 that this only works when q is sufficiently close to p, I think it should be made even clearer that almost any estimate with a finite number of samples will *underestimate* Z^2 when q is not sufficiently close. I agree with the authors that this is probably not an issue at the beginning of training -- but I imagine it becomes an issue as p becomes multimodal/peaky towards convergence, when q cannot follow that distribution anymore. Which begs the question: Why would we train an undirected model p, when the training and evaluation method breaks down around the point when the jointly trained and properly normalized proposal distribution q cannot follow it anymore? Nevertheless, I find the approach interesting and the experiments show that it seems to work for small scale dataset. I think the paper could be improved by paying more attention to this aspect and by investigating the behaviour at the point when q cannot track p anymore. -- update -- I would like to thank the authors (and the other reviewers) for their feedback. I'm still concerned that the focus on the upper bound property of the Z^2 estimator creates a false sense of reliability. I tested the estimator empirically: I used a 10 dim Gaussian as a proposal distribution to estimate Z^2 for another 10 dim Gaussian. Both shared their mean, but while p had a std-deviation of 0.2, q had a std.-deviation of 1. Here the ground truth Z^2 is 1 because p is already a properly normalised distribution. I found that the proposed estimator almost always severely *underestimates* Z^2 for most finite sets of samples from q, even though they constructed the estimator to be an upper bound on Z in expectation (code attached). I would therefor encourage the authors to more explicitly state that the proposed estimator will underestimate Z^2, even though it is an upper bound in expectation. Nevertheless, I think the paper is well written, technically correct and contributes an interesting idea. I therefore updated my overall rating to 6 (marginally above acceptance threashold). -- from __future__ import division import tensorflow as tf distributions = tf.contrib.distributions dim_x = 10 # dimensionality of samples n_samples = 1000 # number of samples form proposal q ones= tf.ones(dim_x) p = distributions.Normal(loc=0.*ones, scale=0.2*ones) # target dist q = distributions.Normal(loc=0.*ones, scale=1.0*ones) # proposal dist x = q.sample(sample_shape=(n_samples,)) log_p = tf.reduce_sum(p.log_prob(x), axis=-1) loq_q = tf.reduce_sum(q.log_prob(x), axis=-1) f_samples = tf.to_float(n_samples) z2_est = tf.reduce_logsumexp(2.*log_p - 2.*loq_q) - tf.log(f_samples) sess = tf.InteractiveSession() print "Real log(Z^2) is 0.0 because P is a proper normalized distribution" for i in xrange(10): print " log(Z^2) estimate: %f5.3" % z2_est.eval() --

Reviewer 2



The presented techniques are very interesting, and I believe they are also very useful. However, I am too concerned about the finite-sample performance of Eq (2). So I strongly suggest a detailed and critical discussion on why and when this bound works, particularly in Eq. (7). I think this paper would have impact and much wider audiences if the authors could provide more proofs to increase credibility. Minors: 1. Line 45, it should be Z(\theta) = \int \tilde p(x) dx. Ex[a(x)] usually means \int a(x) p(x) dx; 2. Line 52, the variance of \hat I should be (E[w2]-I2) / n; 3. Line 148, Wrong parametrization of p(a|x); 4. Eq. (5), it should be \theta^T x instead of log(\theta^T x); 5. Line 275, "...to learn to(two) types..."; 6. Line 280, "u \in R^1 0"; 7. The title of Figure 2 is not right.

Reviewer 3



# Overview The paper presents approximate inference methods for learning undirected graphical models. Learning a Markov random field (MRF) p(x) involves computing the partition function -- an intractable integral $Z(\theta) = \int \bar p_\theta(x) dx$ (where $\bar p_\theta(x)$ is the energy function). The authors use a tractable importance distribution $q_\phi(x)$ to derive an upperbound on the partition function. They argue that optimising the parameters of p and q under this bound can be seen as variational inference with \chi^2-divergence (instead of KL). Essentially the proposed methodology circumvents sampling from an MRF (which would require computing the partition function or a form of MCMC) by sampling from an approximation q and using the same approximation to estimate an upperbound on the log partition function of the MRF (which arises in VI's objective). To obtain an approximations q that is potentially multi-modal the authors employ mixture of K components (termed "non-parametric VI" Gershman et al 2012) and auxiliary variables (in the approximation). The paper then presents 3 applications: 1. 5x5 Ising model: the idea behind this toy application is to assess the quality of the approximate partition function since one can compute the exact one. 2. RBM for MNIST: here they model an RBM p(x) over MNIST-digits using their upperbound on partition function to optimise a lowerbound on the likelihood of the data. 3. Variational auto encoder for (binarised) MNIST: here they model p(x|z)p(z) where the prior p(z) is an RBM. The main point of the empirical comparison was to show that the proposed method can be used to fit a variational auto-encoder with an RBM prior. The authors contrast that with a variational auto-encoder (with the standard Gaussian prior) whose approximate posterior is augmented with auxiliary random variables as to be able to model multiple posterior modes. # Qualitative assessment * Quality The paper is mostly technically sound, but I would like to have a few points clarified: 1. You refer to equation just before line 54 as ELBO, but the ELBO would involve computations about the posterior, which is really not the case here. I can see the parallel to the ELBO, but perhaps this should be clarified in the text. 2. In line 91, I find the text a bit misleading: "the variance is zero, and a single sample computes the partition function". If p = q, the bound in Eq (2) is tight and the expectation equals Z(theta)^2, but that does not mean you should be able to compute it with a single sample. 3. Most of the text in Section 3.4 (optimisation) assumes we can simulate x ~ q(x) directly, but that's not really the case when auxiliary variables are employed (where we actually simulate a ~ q(a) and then x ~ q(x|a)). I am curious about how you estimated gradients, did you employ a reparameterisation for q(a)? How did you deal with the KL term from q(a,x) in Section 5.3 where you have a VAE with an RBM prior? 4. Is the gradient in line 159 correct? I think it might be missing a minus. 5. The energy function in line 204 should have been exponentiated, right? 6. Can you clarify why $\log B(\bar p, q)$ in Equation 7 does not affect p's gradient? As far as I can tell, it is a function of p's parameters (\theta). * Clarity A few results could have been derived explicitly: e.g. gradients of upperbound with respect to q in each of q's formulation that the paper explored; similarly, the estimator for the case where q is q(a,x). * Significance The main advance is definitely interesting and I can imagine people will be interested in the possibility of using MRF priors.